

# Analysis of decadal land cover changes and salinization in Urmia Lake Basin using remote sensing techniques

Yusuf Alizade Govarchin Ghale[1*], Metin Baykara[1], Alper Unal[1]

[1]Climate and Sea Sciences Department, Eurasia Institute of Earth Sciences, Istanbul Technical University, Maslak 34469, Istanbul, Turkey

*Correspondence to*: Yusuf Alizade Govarchin Ghale (alizade@itu.edu.tr)

**Abstract.** Urmia Lake located in the north-west of Iran, is one of the largest hyper-saline lakes in the world. In recent years, most of the Urmia Lake have been rendered to unusable lands. Drought and rapid increase in agricultural activities are the most important reasons behind the shrinkage of the Lake. This kind of exploitation with the added salinity from irrigation

occurring over time has caused increased soil salinity in the basin leading up to desertification. Soil salinity research are crucial to understand underlying causes and consequences of the drying Urmia Lake. In this study, we use remote sensing technology and image processing techniques to detect spatio-temporal variability of salt body, salt affected lands, and development of irrigated lands to estimate the extend of salinization in terms of spectral response of satellite images for the Urmia Lake Basin from 1975 to 2016. The results of this study indicate that salt and salty soil areas has increased

dramatically from 1995 to 2014 and more than 5000 km² of Urmia Lake's water surface area was converted to salt or salty soil bodies during recent years. Salinization and desertification progress are not limited to just dried bottom of the Urmia Lake. Although the area of irrigated lands has increased more than two times during the studied period, soil salinity has increased in regions close to Urmia Lake too. This desertification in the basin have potential to be the source of dust storms, which have adverse effects on people's life and climate as well.


**Keywords:** Urmia Lake, Soil Salinity, NDVI, Desertification, Irrigated lands

## 1 Introduction

Soil salinity is an environmental hazard that leads to soil degradation causing agricultural productivity loss, especially in arid and semiarid regions. In Iran's case, salinity is a major agricultural problem mainly due to the use of low quality saline

irrigation water, low rainfall, and high soil surface evaporation. Urmia Lake located in the north-west of Iran, is one of the largest hyper-saline lakes in the world (Hasanzadeh et al., 2012). In recent years, most of the Urmia Lake have been rendered to unusable. Climate change, drought, and rapid increase in agricultural activities (i.e., opening up more than 80,000 wells and building more than 50 dams) (Iran Water Resource Management Company, 2016) are the most important reasons behind the shrinkage of Urmia Lake. This kind of exploitation with the added salinity from irrigation occurring over time has caused

increased soil salinity in the basin leading up to desertification. Changes caused by drying Urmia Lake may lead to spread of diseases, destruction of agricultural lands, and massive damage on economy might leading to mass migration of local people, similar to what has happened in Aral Sea over the past decades (Zarghami, 2011; UNEP, 2012; AghaKouchak et al., 2015).

In a study by Alesheikh et al., (2007), Landsat data and a hybrid technique including band ration and histogram thresholding were used to detect coastline changes in Urmia Lake. In another study, seasonal and annual variations of Urmia Lake region

between 2000 and 2011 were investigated (Sima et al., 2012) by using remotely sensed data. Their results showed decrease of more than 1500 km² of water surface area of Urmia Lake during the 11 year period. In a recent study, Aghakouchak et al.,



(2015) used Landsat data to determine coastline changes of Urmia Lake between 1972 and 2014. They found that the area of Urmia Lake has decreased about 90% from 1972 to 2014, which are in agreement with previous research. Kabiri et al., (2012) used Landsat images from 1995 to 2011 to calculate the water surface area of Urmia Lake. Rokni et al., (2014) analyzed coastline changes of Urmia Lake using Landsat data between 2000 and 2013. Hamzehpour et al., (2014) analyzed

spatial variation of top soil salinity using ground water SAR and sampling data on a grid of 500 m in an area of 5000 ha close to Urmia Lake during autumn of 2009 and spring of 2010. Their results indicated inverse correlation between top soil salinity and distance from the lake. Sima and Tajrishy, (2014) used spatial interpolation methods to analyze the spatial heterogeneity and temporal changes of the physiochemical parameters of Urmia Lake between October of 2009 and July of 2010. Their results indicated seasonal changes of water quality.

Soil salinity research are crucial to understand underlying causes and consequences of the drying Urmia Lake. Considering the lack of sampling data and limitations of field survey studies in Urmia Lake Basin, our aim is to use remote sensing technology and image processing techniques to detect temporal and spatial variability of salt body, salt affected lands, and development of irrigated lands to estimate the extend of salinization in terms of spectral response of satellite images for the ULB. Temporal changes of different land cover types including salt, salty soil, and water bodies were determined from 1975

to 2016. Normalized Difference Vegetation Index (NDVI), soil Salinity Index (SI) and Maximum likelihood classification methods were used to quantify the acceleration of salinization in the Urmia Lake Basin (ULB) for the study period.

## 2 Study area

Urmia Lake is located in the northwest of Iran (N 37.5°, E 45.5°) between West Azerbaijan and East Azerbaijan provinces with a catchment area of 51876 km². It is the largest inland lake of Iran and the second largest hypersaline lake in the world

(Eimanifar and Mohebbi, 2007; Zarghami, 2011; Hasanzadeh et al., 2012). By many aspects of chemistry, sediments, and morphology, Urmia Lake is similar to Great Salt Lake in State of Utah in USA. The lake is divided into two parts: northern and southern parts. These parts are separated by a causeway that has a 1500 m long bridge, which allows little water exchange between two parts (Eimanifar and Mohebbi, 2007; Rezvantalab et al., 2011; Sima and Tajrish, 2013 ). Its continental climate is affected by the mountains around Urmia Lake and air temperature usually ranges between 0 ° C and -

20 ° C in winter and up to 40 ° C in summer. Urmia Lake Basin has an annual average precipitation between 200 mm - 300 mm. (Eimanifar and Mohebbi, 2007). The measured maximum and minimum water surface elevation of Urmia Lake was about 1278.386 m and 1270.168 m in 1995-June and 2014-September, respectively (Iran water resource management company, 2016).

## 3 Materials and Methods

Accumulation of soluble salt in the soil is defined as soil salinity. Remote sensing techniques, either directly or indirectly, are being used to determine soil salinity changes. By using the direct approach, remote sensing techniques applied on terrain surface using crusts or salinity properties of soil. In indirect approach, biophysical characteristics of vegetation types that are affected by salinity are taken into account. Soil salinity can also be identified by electrical conductivity of a solution that is extracted from a water-saturated soil. In agricultural terms, saline soils have an electrical conductivity more than 4 dS

(decismens per meter) at $25°C$ (Al-Khaier, 2003). Many salinity indexes are proposed for Landsat images to determine salt-affected soils (Goossens and Alavi Panah, 2001; Al-Khaier, 2003; Abbas and Khan, 2007; Abdul-Qadir and Benni, 2010; Abbas et al., 2013; Allbed and Kumar, 2013; Ahmed and Al-Khafaji, 2014; Arnous and Green, 2015). Among these the two



most widely used approaches to monitor soil salinity changes using remote sensing data are Normalized Difference Vegetation Index (NDVI), Salinity Index (SI).

$$SI = \frac{\rho Green + \rho Red}{2} \tag{1}$$

$$NDVI = \frac{\rho NIR - \rho Red}{\rho NIR + \rho Red} \tag{2}$$

Three Landsat frames are needed to cover Urmia Lake (maximum area of about 6000 km²), however, at least eight Landsat frames are needed to cover all parts of Urmia Lake Basin (area of 51876 km²). In this study, 97 Landsat satellite images

dated between 1975 and 2016 were used to identify the changes in land cover of Urmia Lake. The steps of data processing are given in Figure 1. Image pre-processing module, which includes band combination, radiometric calibration, atmospheric correction, and geometric correction, were used to provide geometrically accurate ground surface reflectance images. In the next step, three and eight frames are mosaicked into output images to cover Urmia Lake and Urmia Lake Basin, respectively. After the image mosaicking, Salinity Index in Eq.(1), NDVI in Eq.(2) and Maximum Likelihood Classification methods have

been applied to monitor salinization, vegetation and land cover change for ULB (Levin, 1999; Foody and Mathur, 2004; Kiefer et al., 2008). In UL, Maximum Likelihood Classification methods have been applied to monitor land cover change.

Salinity Index (SI) values that are obtained using ground surface reflectance values of green and red bands range between 0 and 1. High SI values show high salt-affected areas while low values indicate less salt-affected areas (Ahmed and Al-Khafaji, 2013). NDVI is derived from ground surface reflectance values of Red and Infrared bands of Landsat images and it

is used to identify vegetated areas. NDVI values range between -1 and 1. In the study area, values less than zero refers to non-vegetated areas while positive values indicate vegetated lands (The details are given in Table 1) (Anderson et al., 2003; Ritter, 2006; Rawashdeh, 2012).

**Table 1: NDVI ranges and land cover types**

| NDVI Ranges | Cover Type |
| --- | --- |
| -1 to 0 | Non-vegetated areas such as cloud and water |
| 0 to 0.1 | Bare soils, desert lands |
| 0.1 to 0.5 | Sparse vegetation |
| 0.5 to 1 | Dense vegetation such as irrigated areas |


As given in Figure 1, in the final step, salinization progress in the study area is analyzed spatio-temporally by classification of SI maps of Urmia Lake Basin with five classes ranging between 0 - 1. It should be noted that there is a difference between band ranges of Landsat_5 TM and Landsat_8 OLI satellite data due to their technical nature. For example, the green band range in Landsat_5 TM is 0.52 – 0.60 micrometers while the green band range in Landsat_8 OLI is 0.53–0.59 micrometers.

Therefore, it is better to compare the SI results of each satellite data with their own instead of with other satellite data. Landsat_5 TM images were compared to each other for the period between 1984 and 2011 while Landsat_8 OLI results for the period between 2013 and 2016. After investigation of salinity changes, the area of salt, salty soil, and water classes in Urmia Lake were determined to identify desertification progress in the lake. The correlation between salinity changes and vegetation changes is analyzed by using comparison of SI and NDVI values of randomly sampling points of different land

cover types. The irrigated areas in the Urmia Lake Basin were extracted using maximum likelihood classification method to identify the development of irrigation in the basin.





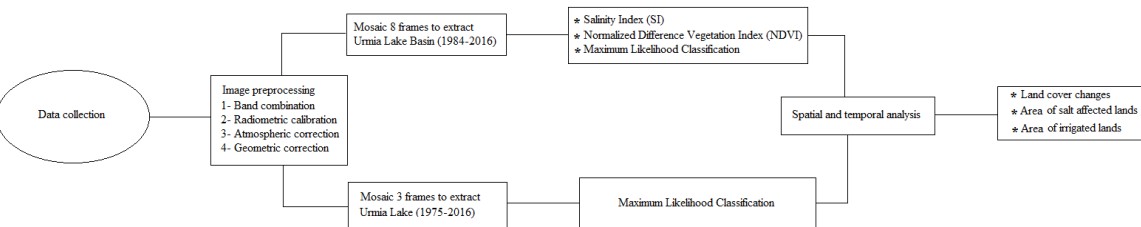

**Figure 1: Image processing steps of quantifying salt affected areas in Urmia Lake Basin**

## 4 Results and discussion

Soil salinization due to irrigation and other intensified agricultural activities, is one of the most severe problems among the many forms of soil degradation (Akramkhanov et al., 2011). Rapid growth in population (Table 2), natural hazards such as drought and exploitation of basin resources are becoming exhaustive. As can be seen in Table 2, urban population near UL, nearly doubled in the last 30 years while rural population is in declining trend. The total population of West Azerbaijan and East Azerbaijan combined has increased during recent decades, however, the point to note here is that the rural population

has decreased over the years while urban population has increased. As a result of this increase in population, agricultural area has increased dramatically.

**Table 2: Population (million) of both West Azerbaijan and East Azerbaijan**

| Year | Urban Population | Rural Population | Total Population |
|------|------------------|------------------|------------------|
| 1985 | 2.5 | 2.5 | 5 |
| 1990 | 2.9 | 2.7 | 5.6 |
| 1995 | 3.3 | 2.5 | 5.8 |
| 2005 | 4.1 | 2.4 | 6.5 |
| 2010 | 4.5 | 2.3 | 6.8 |

As reported in previous studies (Abbas and Khan, 2007, Abdul-Qadir and Benni, 2010; Abbas et al., 2013; Allbed and Kumar, 2013; Arnous and Green, 2015), there exist a strong contrast between normal and salt-affected (salinized) soils in respect to their ground surface conditions. Salt-affected soils have a distinctive feature that make it easier to characterize them; salt efflorescence that had been accumulated over the soil surface by the capillary rise of low quality water. This feature is prominent on top soil and easy to capture in the satellite data. Since salt-affected soils have high spectral

reflectance (Metternichet and Zinck, 2003), especially in the blue band of the visible window their spectral response is higher than normal soils. A field study was conducted where over thousands of ground control points were recorded using GPS while photos of these ground control points were analyzed in order to help classify satellite images in this research. Figure 2 shows the salinization and desertification progress as recorded by sample photos over the years. Water body either dried completely (Figure 2 (b)) or withdrawn from coastal parts to center of the lake (Figure 2 (d)). Shriking water body

leaves dry, arid, saline soils/ground that increases albedo and affecting the climate of the region. In these pictures, different land cover types such as water, salt, salty soil, and soil bodies are visibly distinguishable.




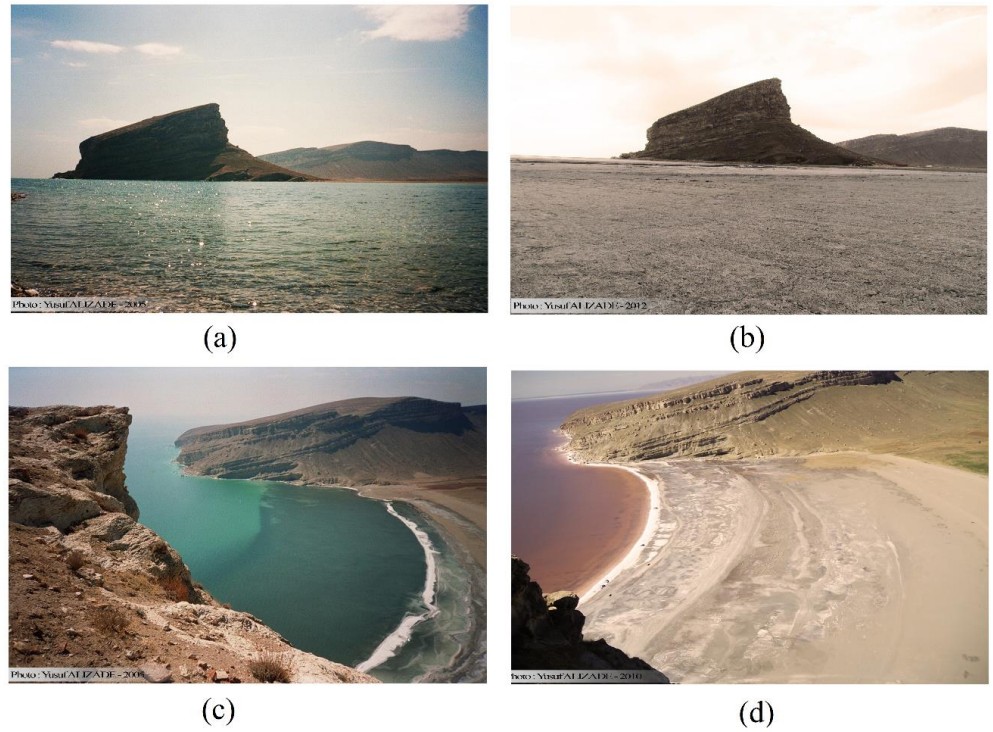

(a)

(b)

(c)

(d)

**Figure 2: Photos of land cover changes in the northwest part of Urmia Lake. (a) Urmia Lake in 2005. (b) Urmia Lake in 2012. (c) Urmia Lake in 2005. (d) Urmia Lake in 2010.**

The soil salinity affects vegetation density and crop growth, and causes desertification. In this work, the corrected satellite
5   images were interpreted using various interpretation keys such as shape, tone, texture, location, and the association. The increased brightness that is easily detectable from the visible part of the spectrum was a clear indicator of the salt present in the soil. Hence, the presence of salt showed higher spectral signals than that of other land features.

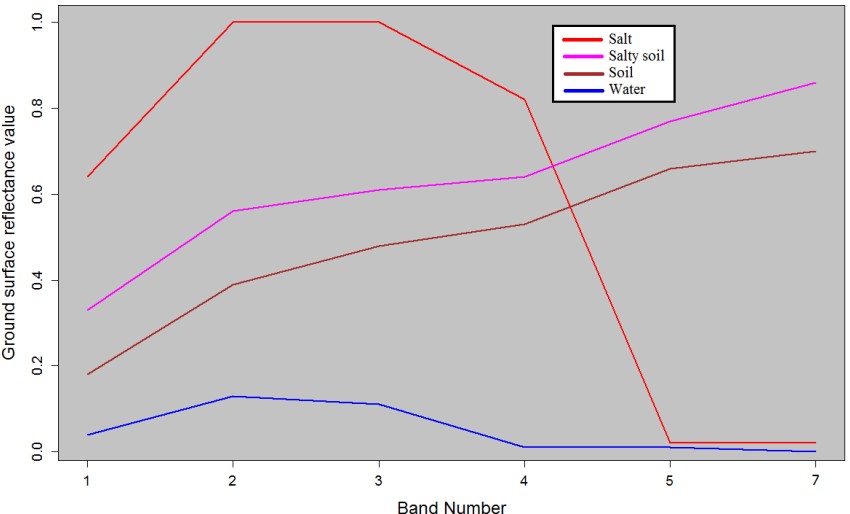

**Figure 3: Spectral profile of salt, salty soil, soil and water bodies in Landsat_5 TM for 2011-August.**



An example of spectral profile of salt, salty soil, soil, and water bodies (one pixel from each body) of Landsat_5 TM for 2011- August is given in Figure 3. It should be noted that these pixels were selected by using ground control points. Spectral profile of different materials can be used to distinguish their spectral characteristics and separate them from each other. According to the figure, the spectral reflectance values of salt and salty soil bodies are much higher than the soil body in the

5   blue (band 1), green (band 2), red (band 3), and NIR (band 4) bands. The spectral reflectance value of water body is lower than other land cover types. Therefore, bands blue, green, red, and NIR can be used to identify soil salinity changes, and salt affected lands.

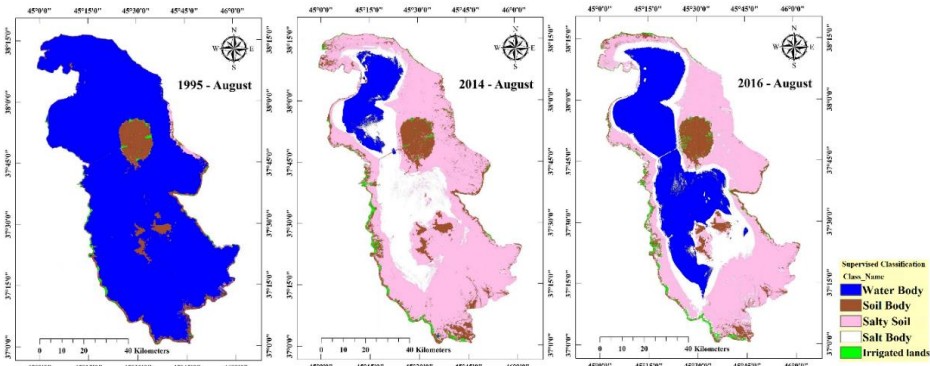

10   **Figure 4: Salinization and desertification progress in Urmia Lake (Classifications are shown for 1995, 2014, and 2016 August.)**

Figure 4 presents progress of desertification over Urmia Lake from 1975 to 2016. White, magenta, blue, and brown colors indicate salt, salty soil, water and soil bodies, respectively. Salt and salty soil bodies have increased considerably from 1995 to 2014. The area of salt and salty soil bodies were about 0 and 63 km², respectively, in 1995 but these values have changed to 1565 and 3711 km², respectively, in 2014. The minimum water surface area was obtained in 2014 with a total value of 586

15   km² whereas maximum water surface area was obtained in 1995 with a total area of 5982 km². After 2014, the process has been reversed. The water surface area has increased about 1490 km² and salt body has decreased about 348 km² in 2016.

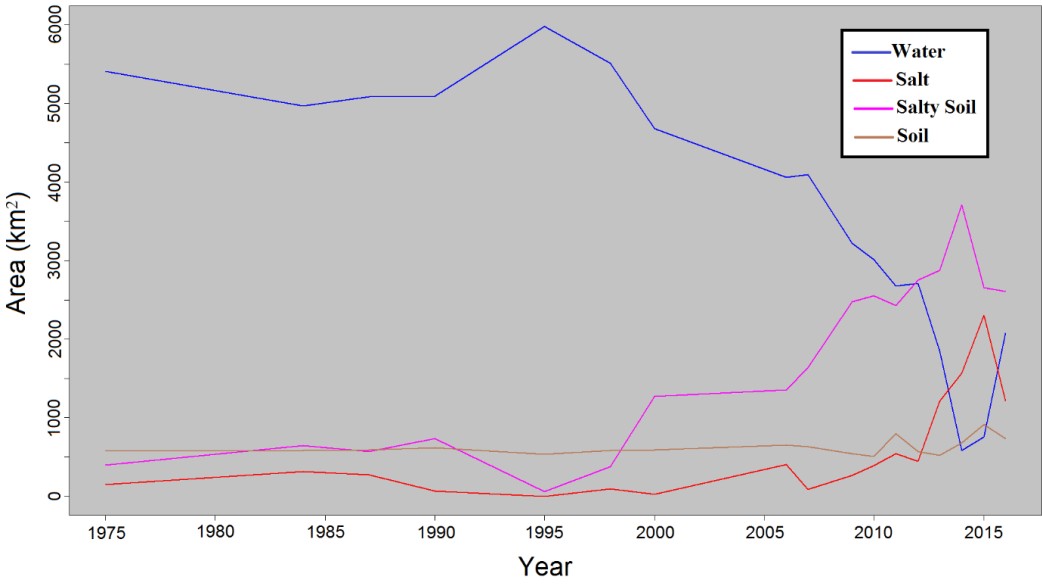




**Figure 5: Land cover changes trend in Summer seaons years from 1975 to 2016**

The trend of land cover changes in summer time of the study period is given in Figure 5. According to this figure, the decrease trend in water surface area and the increase trend in salt and salty soil bodies are considerable after 1995. Water surface area has decreased 32% in 2006 and further up to 90% in 2014 compared to 1995. In addition, salt and salty soil areas have increased dramatically between the same periods. Although, the water surface area has increased after 2014, water surface area of the lake has decreased total of 66% from 1995 to 2016.

Salinization and desertification progress are not limited to just dried bottom of the Urmia Lake. Soil Salinity Index (SI) maps of Urmia Lake Basin derived from Landsat satellite images between 1984 and 2016 (Figure 6 to Figure 8) indicate considerable increase of soil salinity and salt affected lands, especially in the Eastern and Southern parts of Urmia Lake Basin. Some examples of salinization and desertification progress in Urmia Lake Basin (SI maps) from 1984 to 2016 are given in Figure 6 and Figure 7. Green and red colors indicate degree of salt affected lands, green being less and red being more salt affected. Fig.6 indicates SI changes derived from Landsat_5 TM while Figure 7 indicates SI changes derived from Landsat_8 OLI. According to Figure 7, salinization decreased during 2016 compared to 2014 due to increase in the water surface area of Urmia Lake and better climate condition, however, in total the area of high salt affected lands or salt body has increased during recent years.

Table 3 shows the SI classification statistics from 1984 to 2016. The SI values were classified in five classes ranging between 0 - 1. Lower values indicate less salt affected area while higher values mean more salt affected areas. Irrigated lands normally have less SI values ranging 0 – 0.2, and very highly salt affected lands or salt bodies have values in between 0.8 - 1. As it can be seen from the Table 3, the area of highly salt affected lands has increased from 1990 to 2016 (from 0.8% to 3.9%). On the other hand, by increasing the area of irrigated lands, the area of first SI class (0-0.2) has increased too. Soil salinity has increased significantly in all parts of Urmia Lake Basin in 2006 when 32% of water surface area of Urmia Lake dried up compared to that of the Urmia Lake in 1995. Evaporation is strongly dependent on temperature. Therefore, increase in temperature has increased the evaporation too and in turn evaporation caused an increase in soil salinity.

**Table 3: The area of each Soil Salinity Index class and water surface area of Urmia Lake from 1984 to 2016**

| Year | Range (0-0.2) (%) | Range (0.2-0.4) (%) | Range (0.4-0.6) (%) | Range (0.6-0.8) (%) | Range (0.8-1) (%) | Urmia Lake (%) |
|---|---|---|---|---|---|---|
| 1984 | 10.6 | 54.8 | 23.5 | 1.3 | 0.5 | 9.3 |
| 1990 | 18 | 56.4 | 15.3 | 0.7 | 0.1 | 9.5 |
| 2000 | 9.1 | 63.2 | 17.1 | 1.5 | 0.4 | 8.7 |
| 2006 | 5 | 37.3 | 44.6 | 4.7 | 0.8 | 7.6 |
| 2011 | 18.2 | 57.8 | 15 | 2.6 | 1.4 | 5 |
| 2013 | 15.5 | 66.6 | 10.3 | 2.1 | 2 | 3.5 |
| 2014 | 14.1 | 68.5 | 12.6 | 0.6 | 3.1 | 1.1 |
| 2016 | 19.6 | 60.6 | 12 | 1.7 | 2.2 | 3.9 |

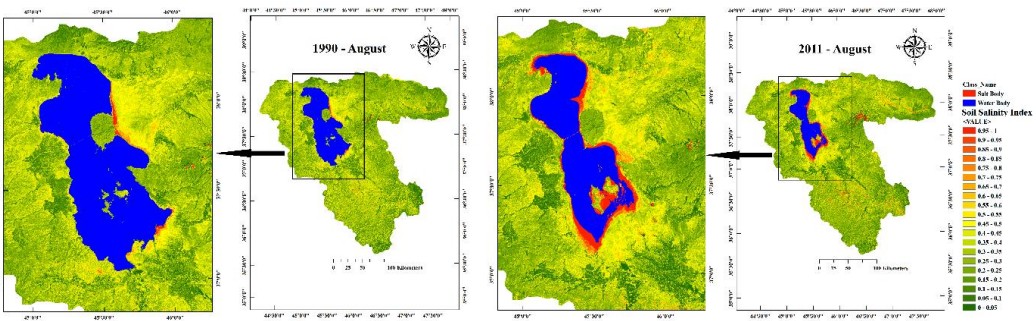





**Figure 6: Water body and Salinity changes in Urmia Lake throughout the years of 1990 – 2011**

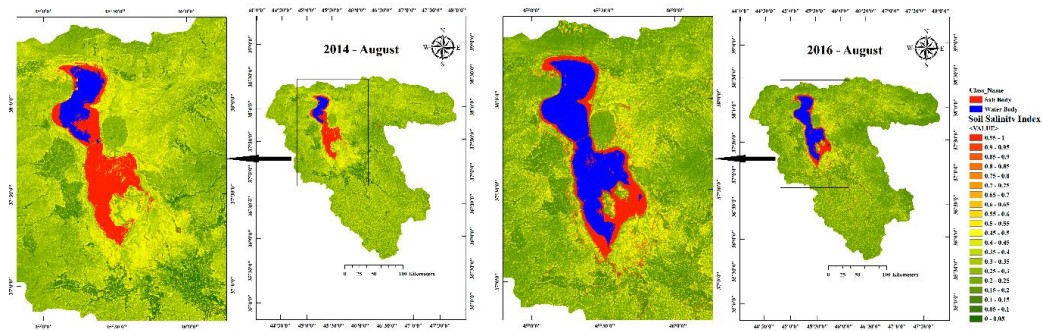

**Figure 7: Water body and Salinity changes in Urmia Lake throughout the years of 2014- 2016**

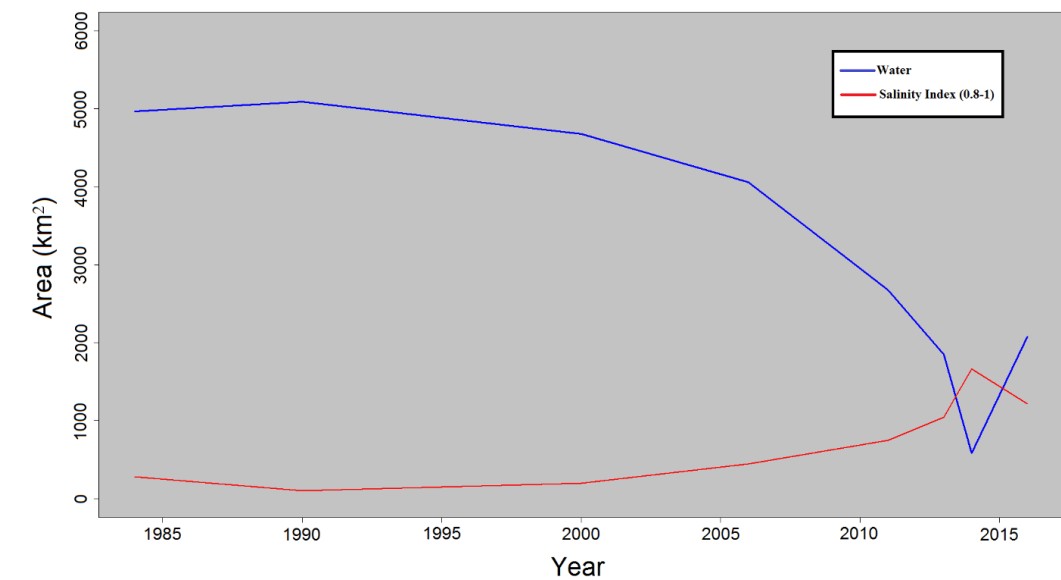

**Figure 8: Inverse correlation between water surface area and area of high salt affected lands.**

10   Figure 8 shows the inverse correlation between shrinking Urmia Lake and increasing area of very high SI values ranging 0.8-
1 from 1984 to 2016. Even though the area of irrigated lands has increased during the recent years, soil salinity has also
increased in agricultural lands located in the regions close to Urmia Lake. Soil salinity affects biomass production in both
arable and irrigated lands. The NDVI and SI values of about 250 sample pixels from different land cover types such as
arable, irrigated and urban lands were randomly selected to quantify the correlation between soil salinity changes and
15   biomass production. The locations of these randomly selected sample pixels can be seen in Figure 9.





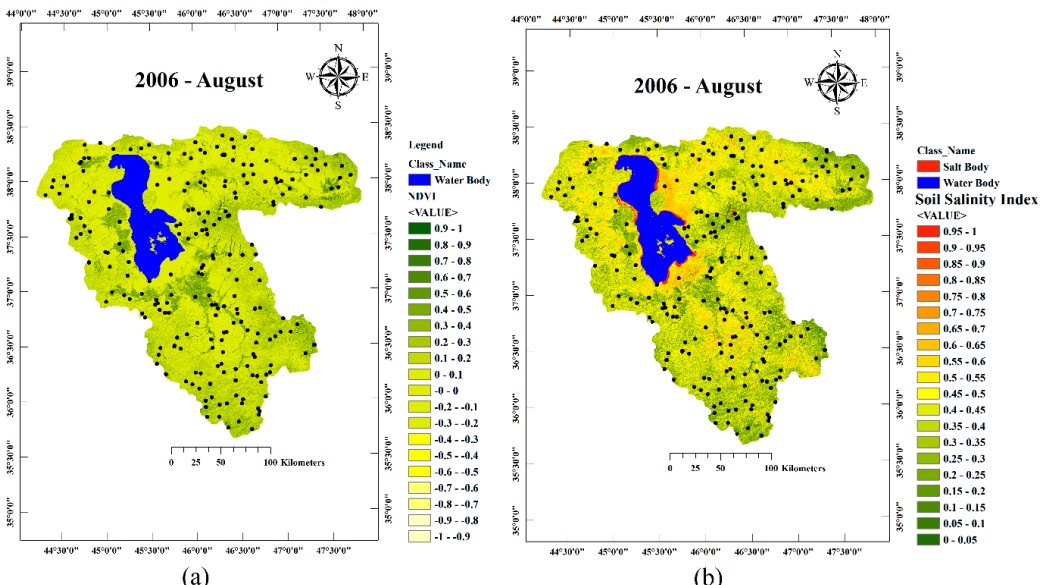

(a)                                    (b)

**Figure 9: Distribution of randomly selected sample pixels to indicate correlation between SI and NDVI changes in 2006**

The correlation between SI and NDVI values of selected sample pixels in 2006 are given in Figure 10. According to this

5    figure, there is an inverse correlation between SI and NDVI changes. NDVI values has decreased while Soil Salinity Index

values increased. Therefore, it can be said that salinization affects the biomass production in all land cover types of Urmia

Lake Basin.

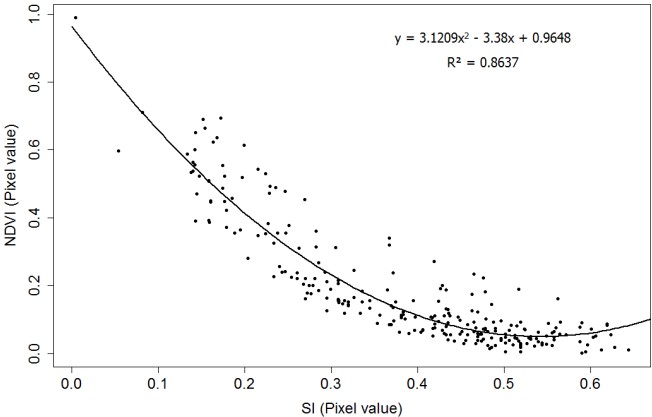

10    **Figure 10: Inverse correlation between SI and NDVI values of randomly sample pixels in 2006**

It is expected to observe a decrease in the size of agricultural areas, especially in the surrounding area of the lake, however,

real life applications show agricultural areas and activities growing rapidly still after all these years.

Figure 11 shows example maps of NDVI changes during studied period. The irrigated lands has increased in all parts of the

15    Urmia Lake Basin. Maximum likelihood supervised classification method was applied to quantify the area of irrigated lands

in Urmia Lake Basin. In order to obtain detailed information about irrigation progress in the Urmia Lake Basin, basin is

divided into three parts and the area of irrigated lands is calculated for western, southern, and eastern parts, independently.





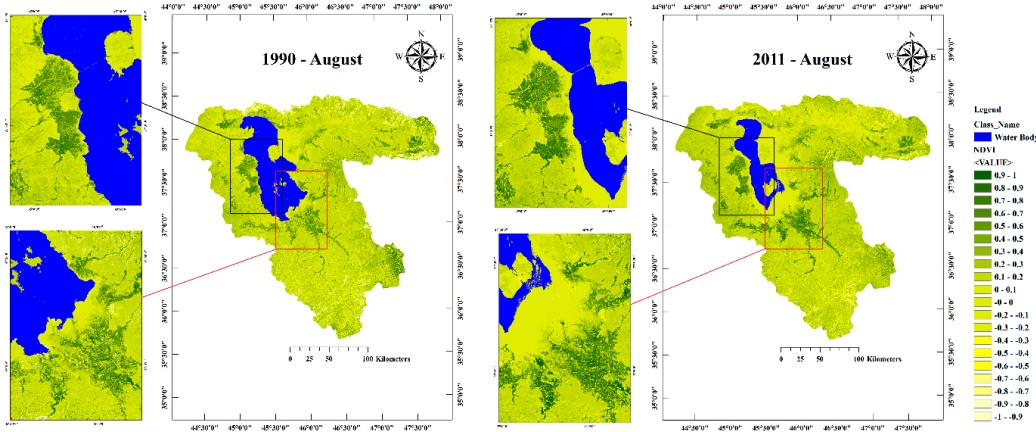

**Figure 11: Water body and NDVI changes in Urmia Lake throughout the years of 1990 – 2011**

5      The results of classification of Urmia Lake Basin to extract irrigation progress in Urmia Lake Basin from 1984 to 2016 are given in Table 4. The irrigated lands has rapidly increased in all parts of Urmia Lake Basin during recent years especially after 2000. The minimum and maximum area of irrigated lands were about 2637 and 5525 km$^2$, respectively, in 1990 and 2011. Increase in the area of irrigated lands is doubled between 1990 - 2011. The cultivation area where is dams are being using, was about 1700 km$^2$ in 2011 (Iran Water Resource Management Company, 2016). Therefore, wells and surface water

10     resources irrigated about 3800 km$^2$ of remaining irrigated lands. The area of irrigated lands has decreased after 2013 about 1000 km$^2$.

| Table 4: Area of Irrigated Lands in Urmia Lake Basin | | | |
|---|---|---|---|
| Year | Area of Irrigated lands in West Part of ULB (Km$^2$) | Area of Irrigated lands in South Part of ULB (Km$^2$) | Area of Irrigated lands in East Part of ULB (Km$^2$) | Total Area of Irrigated Lands in ULB (Km$^2$) |
| 1984 | 767 | 961 | 1127 | 2855 |
| 1990 | 653 | 944 | 1040 | 2637 |
| 2000 | 1026 | 1628 | 1654 | 4308 |
| 2006 | 922 | 1635 | 1486 | 4043 |
| 2011 | 1335 | 2295 | 1895 | 5525 |
| 2013 | 1317 | 2126 | 1963 | 5406 |
| 2014 | 1143 | 1811 | 1687 | 4641 |
| 2016 | 1145 | 1850 | 1550 | 4545 |



The database of agricultural ministry of Iran was used to verify the results obtained in relation to changes in the area of irrigated lands in this study. Figure 12 indicates the total area of irrigated lands of both West Azerbaijan and East Azerbaijan provinces from 2003 to 2014. Even though the area of Urmia Lake Basin is not equal to the area of West Azerbaijan and East Azerbaijan provinces combined, about 90% of the Urmia Lake Basin is located in these provinces. Therefore, most of

the irrigated lands existing in these two provinces are located in Urmia Lake Basin because of this. According to Figure 12 and Table 4 there is a good match between results of our study and the reports of agriculutural ministry of Iran in available years. For example, the decrease trend in the area of irrigated lands in Figure 12 and Table 4 are similar to each other. The area of irrigated lands in two neighbour provinces of Iran was about 5590 km² in 2011, and the area of irrigated lands obtained in this study was about 5525 in the mentioned year. According to the annual reports of agricultural ministry of Iran,

the total production has increased in Urmia Lake Basin due to increasing area of both irrigated and arable lands. According to the results of this study in Table 4, development of agricultural practices in West Azerbaijan is higher than other neighbour provinces of Urmia Lake. At the same time the most input water to Urmia Lake comes from rivers located in the West Azerbaijan. Development of irrigated lands and water mismanagement is one of the reasons behind drying Urmia Lake. Aral sea is a good example to understand and interpret the reasons and effects of drying Urmia Lake. The Aral sea has been

extensively shrunk over the last decades, largely due to water usage from the Amu Darya and Syr Darya rivers for land irrigation (Cretaux et al., 2013). Aralkum region which is a new desert caused by the drying Aral sea is the source of dust storms which have adverse effects on people's life and climate as well ( Indoitu et al., 2015).

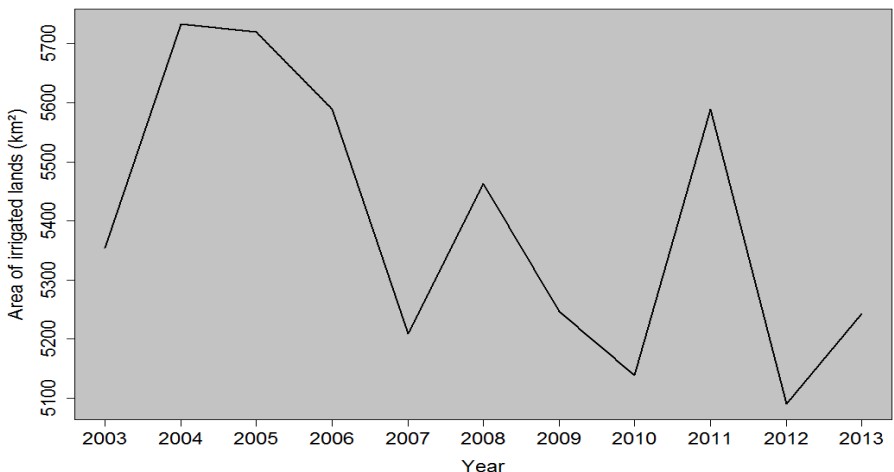

**Figure 12: Total area of irrigated lands in West Azerbaijan and East Azerbijan provinces**

**5 Conclusions**

Mismanagement of surrounding area of Urmia Lake not only results in water being wasted, but also in degradation of soil too. Soil salinization known to cause massive losses of agricultural productivity and damages regional economies. Since groundwater contributes significantly to salt built-up in the surface soil, it is vital to establish monitoring systems (Abbas and

Khan, 2007). In this study, satellite images (data) were used to determine the changes in the soil salinity, area of the water body, and environmentally degraded lands, especially agricultural areas, in Urmia Lake. Soil salinity in irrigated areas is becoming a serious problem for agriculture. Saline soil conditions have resulted in reduction of the value and productivity of considerable areas of land throughout the world (Elhag, 2016). Salinity commonly occurs in irrigated soils due to the





accumulations of soluble salts resulted from continuous use of irrigation waters containing high or medium quantity of dissolved salts (Jingwei et al., 2008; Allbed and Kumar, 2013).

Satellite data showed that salt-affected soils reflect more incident energy in comparison with normal soils and other land use classes in the visible and near-infrared spectrum. Soil salinization and desertification due to drying Urmia Lake and dust

emitted from the dried bottom of the lake is known as one of the most significant problems accelerating desertification in the northwest of Iran. It causes a reduction in biomass production and loss of soil productivity. The occurrence pattern of soil salinity changes from 1975 to 2016, using the image processing techniques, was found to be increased in all parts of Urmia Lake Basin even in agricultural lands. The water surface area of Urmia Lake decreased from 5982 km² in 1995 to 4058 km² in 2006. In other words, slightly over one third of Urmia Lake dried up during the period of 1995 until 2006. Nearly 90% of

the total area dried up by 2014. The water surface area has increased after 2014 due to increase in precipitation and water released from some dams into the lake. Salt and salty soil areas has increased from 1995 to 2014 and more than 5000 km² of Urmia Lake's water surface area was converted to salt or salty soil bodies during recent years. Although the area of irrigated lands has increased from 2637 km2 to 5525 km2 during 20 year period from 1990 to 2011, the soil salinity has increased in agricultural lands close to Urmia Lake. The area of irrigated lands has decreased after 2013 about 1000 km2. The reasons

behind the decrease in the area of irrigated lands after 2013 is an interesting topic and should be investigated.

Salt affected lands has increased in Urmia Lake Basin after 1995 due to drying Urmia Lake, drought, and urbanization. Inverse correlation between shrinking Urmia Lake and expanding area of salt and salt affected lands is concerning. The interactive behavior of salt-affected soils shows that the saline-sodic soils largely prevailed than other known categories in the region. The use of poor quality groundwater, seepage losses from irrigation supply system, poor resource management

are affecting ULB and its surrounding region in a way that the desertification may become a permanent issue. However, inverse changes in water surface area of Urmia Lake and soil salinity are considerable during 2016. The water surface area of Urmia Lake has increased after 2014 due to increasing inflow water to the lake because of water policy changes and increase in precipitation. A simultaneous monitoring project using Remote Sensing technology in conjunction with field survey is recommended to monitor soil salinity and vegetation changes in Urmia Lake Basin. In addition, Geographical

Information Systems (GIS) could be effectively used to conduct spatial and temporal analysis within Urmia Lake and its basin in order to support the decision making process to help and restore Urmia Lake to its ecological level. GIS can also be used to combine different types of data through using models. It can help mix attribute data with spatial and non-spatial data and use them as reliable information in the monitoring and restoring of Urmia Lake.






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
