# Peer review of "Analysis of decadal land cover changes and salinization in Urmia Lake Basin using remote sensing techniques"

_Natural Hazards and Earth System Sciences, 2017_

## Referee Comment (RC1) · Anonymous Referee #1 · 20 Jul 2017

1- What are the novelties of paper in compare to these references? Please describe in the paper/ Fathian F, AD Prasad, Z Dehghan, S Eslamian 2015, Influence of land use/land cover change on land surface temperature using RS and GIS techniques, International Journal of Hydrology Science

Fathian F, S Morid, E Kahya 2015 Identification of trends in hydrological and climatic variables in Urmia Lake basin, Iran, Theoretical and Applied Climatology.

2- In section 2, what is the lake area itself, and nor the basin area. Based on my knowledge the lake area is 5000 km2. Then how it is said in the abstract that the dried area is 5000 km2? All the lake area is dried? 3- Beneath the formulas 1 and 2, describe

the parameters uses in them. 4- The flowchart in figure 1 is not clear. Please enlarge the fonts and improve the graphics. 5- In table 2, the population is for the part of each province which falls in the Urmia lake basin area? Or it is the total population? In the first case some part of the Kurdistan province also falls in the lake basin and it should also mentioned. 6- What is the reference for Table 1. 7- The graphics of the figures 3 and 5 has problem in black and white print. 8- The figures 6-7 are provided I a very small sizes and then difficult to understand. 9- Page 11, line 10, the names of ministries should be written by capitals. 10- In conclusion, first line, please revise accordingly: management of water resources in Urmia Lake basin . . .. 11- The English of the paper should be polished. 12- In discussion I could not find clearly on the causes/drivers of expanding agriculture around the lake. In addition I do not see any citation from the Urmia Lake Restoration Program. What are their interactive responses to the findings of tis valuable study?

---

## Author Comment (AC1) · 22 Aug 2017

Dear Dr. Rosa Lasaponara, Editor Journal of Natural Hazards and Earth System Sciences We would like to thank you and all the reviewers for the valuable comments that improved the quality of our paper. The paper has been thoroughly and carefully revised according to the reviewers' comments. Below are the responses given to the comments raised by the reviewer. Sincerely, ******** Reviewer #1: Comment 1- What are the novelties of paper in compare to these references? Please describe in the paper/ Fathian F, AD Prasad, Z Dehghan, S Eslamian 2015, Influence of land use/land cover change on land surface temperature using RS and GIS techniques, International

Journal of Hydrology Science. 2- Fathian F, S Morid, E Kahya 2015 Identification of trends in hydrological and climatic variables in Urmia Lake basin, Iran, Theoretical and Applied Climatology. Author's Respond: Recent studies on Urmia Lake and its basin, mainly focused on the coastline, water surface area changes, and partial analysis while in this study temporal changes of different land cover types (such as salt, salty soil, and water body) were analyzed between 1975 and 2016. The area of irrigated lands in Urmia Lake Basin were determined between 1984 and 2016 to better understand how human activities have changed water consumption trends in the basin. Land field survey data have been used in order to improve the accuracy of the findings. Desertification progress in the ULB is determined after salinity changes, the area of salt, salty soil, and water classes in Urmia Lake were identified. The results of this research can play an important role in management of the northwest of Iran by providing accurate information on changes that have been happening in ULB in the last 41 years. We believe this study has the potential to be used as a reference study in related salinization and desertification studies. We have included the suggested references in the Introduction part (page 1, lines 29-30 and page 2, lines 11-17). Page 1, lines 29-30: Additionally, impact of increase in temperature and overuse of water resources on the drying Urmia Lake was investigated in a recent study (Fathian et al., 2015). Page 2, Lines 11-17 Fathian et al., (2015) used three Landsat images to analyze Land use/Land cover (LULC) changes and their impacts on Land Surface Temperature (LST) in the east part of Urmia Lake Basin (ULB) in 1989, 2002, and 2011. Results indicated that human activities and urbanization had affected the vegetation types and LST from the high temperature-sparse vegetation to the low temperature-dense vegetation during the period between 1989-2002 and from low temperature-dense vegetation to the high temperature-dense vegetation in period between 2002-2011. In previous studies a lot of valuable investigations have been conducted on Urmia Lake and its basin but there are no detailed studies in long term salinization progress and development of irrigated lands in all parts of ULB. We have explained the novelty of paper in compare to previous studies in page 2, in lines 15 to 25: In previous studies a lot of valuable

investigations have been conducted on Urmia Lake and its basin but there are no detailed studies in long term salinization progress and development of irrigated lands in all parts of ULB. Soil salinity research are crucial to understand the underlying causes and consequences of the drying Urmia Lake. Considering the lack of sampling data and limitations of field survey studies in Urmia Lake Basin, our aim is to use remote sensing technology and image processing techniques to detect temporal and spatial variability of salt body, salt affected lands, and development of irrigated lands to estimate the extend of salinization in terms of spectral response of satellite images for the ULB. Temporal changes of different land cover types including salt, salty soil, and water bodies were determined from 1975 to 2016. Normalized Difference Vegetation Index (NDVI), soil Salinity Index (SI) and Maximum likelihood classification methods were used to quantify the acceleration of salinization in the Urmia Lake Basin (ULB) for the study period.

Comment 2- In section 2, what is the lake area itself, and nor the basin area. Based on my knowledge the lake area is 5000 km2. Then how it is said in the abstract that the dried area is 5000 km2? All the lake area is dried? Reply: We have estimated the lake area as 5982 km2 for 1995, the maximum value for the period between 1972 and 2016. Similar number is given in the literature by Eimanifar and Mohebbi, 2007; Zarghami, 2011; Hasanzadeh et al., 2012. The following sentences were added to clarify this point on page 2 (lines 27-30), page 2 (lines . . .), and page 6 (lines . . .): Page 2, lines 27-30: Urmia Lake which has a surface area ranging between 5000 and 6000 km2 is located in the northwest of Iran (N 37.5°, E 45.5°) between West Azerbaijan and East Azerbaijan provinces with a catchment area of 51876 km$^2$. It is the largest inland lake of Iran and the second largest hypersaline lake in the world (Eimanifar and Mohebbi, 2007; Zarghami, 2011; Hasanzadeh et al., 2012). Page 6, lines 14-18: The results of this study indicate that salt and salty soil areas has increased dramatically from 1995 to 2014 and more than 5000 km$^2$ of Urmia Lake's water surface area was turned into salt or salty soil bodies during this period. The minimum water surface area was estimated for 2014 with a total value of 586 km2 whereas maximum water surface

area was determined for 1995 with a total area of 5982 km2. The water surface area has increased to 1490 km2 and salt body has decreased about 348 km2 in 2016.

Comment 3- Beneath the formulas 1 and 2, describe the parameters uses in them. Reply: We have described the formulas 1 and 2 on page 3, lines 12-16: SI= (Green+ Red)/2 (1) where, Green and Red are ground surface reflectance values of green and red bands respectively.

NDVI=(NIR- Red)/(NIR+ Red) (2) where, NIR and Red are ground surface reflectance values of Near Infrared and red bands respectively.

Comment 4- The flowchart in figure 1 is not clear. Please enlarge the fonts and improve the graphics. Reply: We have improved the quality of Figure 1.

Comment 5- In table 2, the population is for the part of each province which falls in the Urmia lake basin area? Or it is the total population? In the first case some part of the Kurdistan province also falls in the lake basin and it should also mentioned. Reply: Table 2 includes the total population of just West Azerbaijan and East Azerbaijan (Population in West Azerbaijan + Population in East Azerbaijan). We have used the data of these provinces because about 90% of Urmia Lake Basin is located in these provinces and they are located in neighboring of Urmia Lake.

Comment 6- What is the reference for Table 1. Reply: The references are given in the text (on page 3, lines 29-30): Anderson et al., 2003; Ritter, 2006; Rawashdeh, 2012, USGS, 2016).

We have cited 4 references (3 references in last version) about NDVI, in page 3, lines 30-31. The fourths references is related to the USGS (https://phenology.cr.usgs.gov/ndvi_foundation.php). For more information you can visit this URL too. http://www.pvts.net/pdfs/ndvi/3_3_ndvi.PDF

Comment 7- The graphics of the figures 3 and 5 has problem in black and white print. Reply: We have improved the quality of these figures.

Comment 8- The figures 6-7 are provided in a very small sizes and then difficult to understand. Reply: We have improved the quality of these figures.

Comment 9- Page 11, line 10, the names of ministries should be written by capitals. Reply: We have revised the names of ministries.

Comment 10- In conclusion, first line, please revise accordingly: management of water resources in Urmia Lake basin: : :. Reply: We have revised this part accordingly.

Comment 11- The English of the paper should be polished. Reply: English of the paper has been improved. Repetitive parts are removed, and general flow of the manuscript has been improved.

Comment 12- In discussion I could not find clearly on the causes/drivers of expanding agriculture around the lake. In addition I do not see any citation from the Urmia Lake Restoration Program. What are their interactive responses to the findings of this valuable study? Reply: The question asked by the reviewer is beyond the scope of this study. Possible logical reasons behind the expansion of agricultural areas are the increasing population and their needs and agricultural policies. Unfortunately, our objectives do not include finding the reasons behind the increasing trend in agricultural activities. We don't have enough scientific and technical information about the plans and activities "Urmia Lake Restoration Program". However, it is possible that the increase after 2014 might be contributed to these restoration programs. We follow this program through: http://ulrp.sharif.ir/en. We have plan to share our findings with them through publication of this paper.

We have added the revised paper to the supplement part of this page.

Please also note the supplement to this comment:
https://www.nat-hazards-earth-syst-sci-discuss.net/nhess-2017-212/nhess-2017-212-AC1-supplement.pdf

Mosaic 8 fra
Urmia Lake

Image preprocessing
1- Band combination
2- Radiometric calibration
3- Atmospheric correction
4- Geometric correction

Data collection

Mosaic 3 fra
Urmia Lake

Fig. 1.

[Figure]

[Figure]

[Figure]

[Figure]

[Figure]

45°0'0"

[Figure]

45°0'0"

[Figure]

[Figure]

44°0'0" 44°30'0" 45°0'0" 45

39°0'0"

[Figure]

y = 3.1209x² - 3.[...]

R² = 0.8[...]

NDVI (Pixel value)

[Figure]

[Figure]

**Supplement:**

**Analysis of decadal land cover changes and salinization in Urmia Lake Basin using remote sensing techniques**

Yusuf Alizade Govarchin Ghale[1*], Metin Baykara[1], Alper Unal[1]

[1]Climate and Sea Sciences Department, Eurasia Institute of Earth Sciences, Istanbul Technical University, Maslak 34469, Istanbul, Turkey

[*]*Correspondence to*: Yusuf Alizade Govarchin Ghale (alizade@itu.edu.tr)

**Abstract.** Urmia Lake, located in the north-west of Iran, is one of the largest hyper-saline lakes in the world. In recent years, most of the Urmia Lake has been rendered to unusable lands. Drought and rapid increase in agricultural activities are the most important reasons behind the shrinkage of the lake. This kind of exploitation with the added salinity from irrigation occurring over time has caused increased soil salinity in the basin leading up to desertification. Soil salinity research are crucial to understand underlying causes and consequences of the drying Urmia Lake. In this study, we use remote sensing technology and image processing techniques to detect spatio-temporal variability of salt body, salt affected lands, and development of irrigated lands to estimate the extend of salinization in terms of spectral response of satellite images for the Urmia Lake Basin from 1975 to 2016. The results of this study indicate that salt and salty soil areas has increased dramatically from 1995 to 2014 and more than 5000 km² of Urmia Lake's water surface area was converted to salt or salty soil bodies during this period. Salinization and desertification progresses are not limited to just dried bottom of the Urmia Lake. Although the area of irrigated lands has increased more than two times during the studied period, soil salinity has increased in regions close to Urmia Lake. This desertification in the basin have potential to be the source of dust storms, which have adverse effects on people's life and climate as well.

**Keywords:** Urmia Lake, Soil Salinity, NDVI, Desertification, Irrigated lands

**1 Introduction**

Soil salinity is an environmental hazard that leads to soil degradation causing agricultural productivity loss, especially in arid and semiarid regions. In Iran's case, salinity is a major agricultural problem mainly due to the use of low quality saline irrigation water, low rainfall, and high soil surface evaporation. Urmia Lake located in the north-west of Iran, is one of the largest hyper-saline lakes in the world (Hasanzadeh et al., 2012). In recent years, most of the Urmia Lake has been rendered to unusable. Drought and rapid increase in agricultural activities (i.e., opening up more than 80,000 wells and building more than 50 dams) (Iran Water Resource Management Company, 2016) are the most important reasons behind the shrinkage of Urmia Lake. Additionally, impact of increase in temperature and overuse of water resources on the drying Urmia Lake was investigated in a recent study (Fathian et al., 2015). Exploitation of the lake area with the added salinity from irrigation occurring over time has caused increased soil salinity in the basin leading up to desertification. Environmental changes caused by drying Urmia Lake may lead to spread of diseases, destruction of agricultural lands, and massive damage on economy resulting in mass migration of local people, similar to what has happened in Aral Sea over the past decades (Zarghami, 2011; UNEP, 2012; AghaKouchak et al., 2015). In a study by Alesheikh et al., (2007), Landsat data and a hybrid technique including band ration and histogram thresholding were used to detect coastline changes in Urmia Lake. In another study, seasonal and annual variations of Urmia Lake region between 2000 and 2011 were investigated (Sima et al., 2012) by

using remotely sensed data. Results of this study showed a decrease of more than 1500 km$^2$ of water surface area of during the 11-year period. In a recent study, Aghakouchak et al., (2015) used Landsat data to determine coastline changes of Urmia Lake between 1972 and 2014. According to the results that the area of Urmia Lake has decreased about 90% from 1972 to 2014, which is in agreement with previous research. Kabiri et al., (2012) used Landsat images from 1995 to 2011 to calculate the water surface area of Urmia Lake. Rokni et al., (2014) analyzed coastline changes of Urmia Lake using Landsat data between 2000 and 2013. Hamzehpour et al., (2014) analyzed spatial variation of top soil salinity using ground water SAR and sampling data on a grid of 500 m in an area of 5000 ha close to Urmia Lake during autumn of 2009 and spring of 2010. Their results indicated inverse correlation between top soil salinity and distance from the lake. Sima and Tajrishy, (2014) used spatial interpolation methods to analyze the spatial heterogeneity and temporal changes of the physiochemical parameters of Urmia Lake between October of 2009 and July of 2010, results indicated seasonal changes of water quality. Fathian et al., (2015) used three Landsat images to analyze Land use/Land cover (LULC) changes and their impacts on Land Surface Temperature (LST) in the east part of Urmia Lake Basin (ULB) in 1989, 2002, and 2011. Results indicated that human activities and urbanization had affected the vegetation types and LST from the high temperature-sparse vegetation to the low temperature-dense vegetation during the period between 1989-2002 and from low temperature-dense vegetation to the high temperature-dense vegetation in period between 2002-2011. In previous studies a lot of valuable investigations have been conducted on Urmia Lake and its basin but there are no detailed studies in long term salinization progress and development of irrigated lands in all parts of ULB.

Soil salinity research are crucial to understand the underlying causes and consequences of the drying Urmia Lake. Considering the lack of sampling data and limitations of field survey studies in Urmia Lake Basin, our aim is to use remote sensing technology and image processing techniques to detect temporal and spatial variability of salt body, salt affected lands, and development of irrigated lands to estimate the extend of salinization in terms of spectral response of satellite images for the ULB. Temporal changes of different land cover types including salt, salty soil, and water bodies were determined from 1975 to 2016. Normalized Difference Vegetation Index (NDVI), soil Salinity Index (SI) and Maximum likelihood classification methods were used to quantify the acceleration of salinization in the Urmia Lake Basin (ULB) for the study period.

**2 Study area**

Urmia Lake which has a surface area ranging between 5000 and 6000 km2 is located in the northwest of Iran (N 37.5°, E 45.5°) between West Azerbaijan and East Azerbaijan provinces with a catchment area of 51876 km². It is the largest inland lake of Iran and the second largest hypersaline lake in the world (Eimanifar and Mohebbi, 2007; Zarghami, 2011; Hasanzadeh et al., 2012). By many aspects of chemistry, sediments, and morphology, Urmia Lake is similar to Great Salt Lake in State of Utah in USA. The lake is divided into two parts: northern and southern parts. These parts are separated by a causeway that has a 1500 m long bridge, which allows little water exchange between two parts (Eimanifar and Mohebbi, 2007; Rezvantalab et al., 2011; Sima and Tajrish, 2013). Its continental climate is affected by the mountains around Urmia Lake and air temperature usually ranges between 0 º C and -20 º C in winter and up to 40 º C in summer. Urmia Lake Basin has an annual average precipitation between 200 mm - 300 mm. (Eimanifar and Mohebbi, 2007). The measured maximum and minimum water surface elevation of Urmia Lake was about 1278.386 m and 1270.168 m in 1995-June and 2014-September, respectively (Iran water resource management company, 2016).

**3 Materials and Methods**

Accumulation of soluble salt in the soil is defined as soil salinity. Remote sensing techniques, either directly or indirectly, are being used to determine soil salinity changes. By using the direct approach, remote sensing techniques applied on terrain surface using crusts or salinity properties of soil. In indirect approach, biophysical characteristics of vegetation types that are
5 affected by salinity are taken into account. Soil salinity can also be identified by electrical conductivity of a solution that is extracted from a water-saturated soil. In agricultural terms, saline soils have an electrical conductivity more than 4 dS (decismens per meter) at $25^{\circ}$C (Al-Khaier, 2003). Many salinity indexes are proposed for Landsat images to determine salt-affected soils (Goossens and Alavi Panah, 2001; Al-Khaier, 2003; Abbas and Khan, 2007; Abdul-Qadir and Benni, 2010; Abbas et al., 2013; Allbed and Kumar, 2013; Ahmed and Al-Khafaji, 2014; Arnous and Green, 2015). Among these the two
10 most widely used approaches to monitor soil salinity changes using remote sensing data are Normalized Difference Vegetation Index (NDVI), Salinity Index (SI).

$$SI = \frac{\rho Green + \rho Red}{2} \tag{1}$$

where, $\rho Green$ and $\rho Red$ are ground surface reflectance values of green and red bands, respectively.

$$NDVI = \frac{\rho NIR - \rho Red}{\rho NIR + \rho Red} \tag{2}$$

15 where, $\rho NIR$ and $\rho Red$ are ground surface reflectance values of Near Infrared and red bands, respectively.

[revised manuscript text omitted]

Iran Water Resource Management Company (2016) WRM. http://wrm.ir/. Accessed 15 March 2016

United States Geological Survey (2016) EarthExplorer. https://earthexplorer.usgs.gov/. Accessed 15 August 2016

United States Geological Survey (2016) EarthExplorer. https://glovis.usgs.gov/. Accessed 20 August 2015

West Azerbaijan Meteorological Office (2016) http://www.met-ag.ir/. Accessed 10 July 2016

East Azerbaijan Meteorological Office (2016) http://www.eamo.ir/. Accessed 15 July 2016

---

## Referee Comment (RC2) · Anonymous Referee #2 · 2 Jan 2018

The manuscript investigates the problem of water decrease and soil salinity increase in the basin of a large lake in northwestern Iran. The investigation is based only on satellite (Landsat) images, in particular in salinity index and vegetation index. These indicators are extrapolated by satellite images over a period ranging between 1975 and 2016. It is emphasized how particularly during the last 20 years the level of the lake dramatically decreased as well as soil salinity increased, thus caused a severe degradation of soil, in terms of farming. The manuscript focuses on an interesting topic, however I think there are some critical conceptual issues which prevent the paper from being published as it is. These are:

[Figure]

1. The proposed methodology is exclusively based on satellite images. An extensive climatic analysis is missing. I think this is a critical point, since, as correctly emphasized by Authors, starting from 2014 something is changing, possibly due to precipitation increase.

2. It is not clear the relation between increase of urban population and lake level decrease. It should be investigated how water distribution systems and water pumping increased, or if new dams were built.

3. It is not clear how the agriculture affected the problem of lake depauperation and soil desertification: did irrigation practices changed? Did the crops changed? Apparently in the latest years agriculture decreased, is this related to a low fertility of soils, or to low availability of water?

4. Almost nothing is mentioned about the hydrogeological and geological features of the catchments, I think these important features, which may help the reader in understanding the investigated problems.

5. The entire manuscript should be proofread, there are a lot of typos and English is sometimes poor.

Given the aforementioned points, I would suggest to return the manuscript to the Authors for a major review.

---

## Author Comment (AC3) · 7 Feb 2018

Dear Prof. Rosa Lasaponara,

Editor

Journal of Natural Hazards and Earth System Sciences

We would like to thank you and to the second reviewer for the valuable comments that improved the quality of our paper. The manuscript has been thoroughly and carefully revised according to the reviewers' comments. Below are the responses given to the comments raised by the reviewer. Sincerely,
* * *
Reviewer #2:

Comment 1- The proposed methodology is exclusively based on satellite images. An extensive climatic analysis is missing. I think this is a critical point, since, as correctly emphasized by Authors, starting from 2014 something is changing, possibly due to precipitation increase.

Author's Respond:

As the referee has pointed out, this study is exclusively based on remote sensing technology and image processing techniques. Earlier, we have done a detailed study on climate's role and main factors behind the drying up of Urmia Lake in our published paper "Investigation Anthropogenic Impacts and Climate Factors on Drying up of Urmia Lake Using Water Budget and Drought Analysis. Alizade Govarchin Ghale, Y., Altunkaynak, A. & Unal, A. Water Resour Manage (2018) 32: 325. https://doi.org/10.1007/s11269-017-1812-5)". In this study, we have found that anthropogenic factors play a crucial role in water level fluctuations of Urmia Lake and their effect on the lake are more significant than the climatic factors. We have included the results of our published paper in the manuscript (page 2, lines 15-20) to summarize the climatic analysis. Moreover, we have explained the reasons behind the increasing water surface area of Urmia Lake on page 14, lines 23-29. Unfortunately, the data of Urmia Lake Basin meteorological stations are available up until 2014, and there is no public access to the data after 2014.

Edits to the manuscript: Page 2, lines 17-21:

Alizade Govarchin Ghale et al., (2018) have done a detailed study on climatic conditions and main factors behind the drying up of Urmia Lake. They have concluded that anthropogenic factors play a crucial role in the water level fluctuations of Urmia Lake and their impact on the lake is more significant than climatic factors. According to their
study results, impact of precipitation and evaporation factors on drying up of Urmia Lake in recent years, 17% and 3%, respectively, and anthropogenic factors account for about 80% of the shrinking Urmia Lake.

Edits to the manuscript: Page 18, Lines 10-15:

The water surface area of Urmia Lake has increased after 2014. The scientific reports of meteorological office of Iran and Urmia Lake Restoration Program indicate that precipitation has increased between 2014 and 2016. In addition, the government claims that they have released more water from dams into the lake to prevent the complete drying of Urmia Lake, so it is possible that both climate factors and anthropogenic factors have played a positive role in restoring of Urmia Lake from 2014 to 2016.

Comment 2- It is not clear the relation between increase of urban population and lake level decrease. It should be investigated how water distribution systems and water pumping increased, or if new dams were built.

Author's Respond:

We have explained the human impact on Urmia Lake in our published paper in Investigation Anthropogenic Impacts and Climate Factors on Drying up of Urmia Lake Using Water Budget and Drought Analysis. Alizade Govarchin Ghale, Y., Altunkaynak, A. & Unal, A. Water Resour Manage (2018) 32: 325. https://doi.org/10.1007/s11269-017-1812-5).

The detailed information and statistics about anthropogenic impacts on Urmia Lake were analyzed in part "3.3 Human impacts on Urmia Lake" of our published paper. We have explained how established dams and wells have affected water distribution system in Urmia Lake Basin and how mismanagement of water resources in the basin has led to gradual drying of Urmia Lake in recent years. The results of this paper estimated a strong dependency (r=0.87) between population and the area of irrigated lands. In the text, we have made the following edits.

[Figure]

Edits to the manuscript: Page 5, lines 4-13:

The northwest part of Iran is one of the agricultural development centers of Iran, and a variety of products, such as apple, wheat, sugar beet, tomato, potato, and grapes are grown in this region. Dams and groundwater resources provide the required water for developing irrigated lands in the basin. There are in total 103 dams in the West Azerbaijan and East Azerbaijan provinces (Iran Water Resource Management Company 2016). Of these dams, 56 are located in Urmia Lake Basin and 42 of them were built between 1990 and 2014. The area of agricultural lands that have been irrigated by dams has been doubled in the last 15 years (Alizade Govarchin Ghale et al., 2018). Ground water resources such as deep wells and semi deep wells are the other sources of irrigation in the study area. The annual discharge water from ground water resources has increased during recent years. The withdrawal water from groundwater resources was about 2156 mcm in 2012 while it was about 702 mcm in 1972. Therefore, dams and wells prevent water from entering the lake and they have a direct effect on the gradual drying of Lake Urmia.

Comment 3- It is not clear how the agriculture affected the problem of lake depaupera-tion and soil desertification: did irrigation practices changed? Did the crops changed? Apparently in the latest years agriculture decreased, is this related to a low fertility of soils, or to low availability of water?

Author's Respond:

There is not a consistent and extensive database of crop types for the study area, but only one report on this subject and it is not sufficient. In order to monitor the crops grown in the region an extensive field campaign is required. The only available data is related to the scientific reports of the Ministry of Agriculture-Jahad from 2003 to 2013 and this data do not give us any information about crops grown before 2000. The water level of Urmia Lake has decreased dramatically after 1995 and we need to have a complete database about crops before this time. Unfortunately, there is no

public access to this database. According to our results, the area of irrigated lands has decreased during recent years and this could be due to low fertility of soil or low availability of water or even this can be due to water policy changes in the basin in order to provide more water for Urmia Lake. The question asked by the reviewer is beyond the scope of this study. We don't have enough scientific and technical information and updated data from dams, wells and other water resources to judge about the reasons behind the decreasing area of agricultural lands in the study area. This is a question that we have raised in the final section of our paper, in page 18, lines 4-5.

Comment 4- Almost nothing is mentioned about the hydrogeological and geological features of the catchments, I think these important features, which may help the reader in under- standing the investigated problems.

Author's Respond:

We have explained the hydrogeological and geological features of the basin in page 3, lines 2-6.

Edits to the manuscript: page 2, lines 32-39 and page 3, lines 1-7:

Urmia Lake has a surface area ranging between 5000 and 6000 km2 and is located in the northwest of Iran (N 37.5°, E 45.5°) between West Azerbaijan and East Azerbaijan provinces with a catchment area of 51876 km$^2$. The size of this catchment area is approximately 3% of entire area of Iran. The catchment contains 21 permanent and 39 episodic rivers (Ghaheri and Baghal-Vayjooee, 1999). It is the largest inland lake of Iran and the second largest hypersaline lake in the world (Eimanifar and Mohebbi, 2007; Zarghami, 2011; Hasanzadeh et al., 2012). Saline lakes are susceptible to environmental changes because their properties vary with the changes occur in their hydrologic budgets (Romero and Melack, 1996). The geology of the area ranges from pre-cambrian to Quaternary (Alipour, 2006). By many aspects of chemistry, sediments, and morphology, Urmia Lake is similar to Great Salt Lake in State of Utah in USA. The lake is divided into two parts: northern and southern parts. These parts are separated

by a causeway that has a 1500 m long bridge, which allows little water exchange be-
tween two parts (Eimanifar and Mohebbi, 2007; Rezvantalab et al., 2011; Sima and
Tajrish, 2013). Its continental climate is affected by the mountains around Urmia Lake
and air temperature usually ranges between 0 $^\circ$ C and -20 $^\circ$ C in winter and up to 40
$^\circ$ C in summer. Urmia Lake Basin has an annual average precipitation between 200
mm - 300 mm. (Eimanifar and Mohebbi, 2007). The measured maximum and minimum
water surface elevation of Urmia Lake was about 1278.386 m and 1270.168 m in 1995-
June and 2014-September, respectively (Iran water resource management company,
2016).

Comment 5- The entire manuscript should be proofread, there are a lot of typos and
English is sometimes poor.

Author's Respond:

English of the paper has been improved. Repetitive parts are removed, and general
flow of the manuscript has been improved.

Please also note the supplement to this comment:
https://www.nat-hazards-earth-syst-sci-discuss.net/nhess-2017-212/nhess-2017-212-
AC3-supplement.pdf
* * *
2017-212, 2017.

[Figure]

**Fig. 1.**

[Figure]

(a)

(b)

(c)

(d)

**Fig. 2.**

**Fig. 3.**

[Figure]

**Fig. 4.**

Fig. 5.

**Fig. 6.**

Interactive
comment

**Fig. 7.**

Fig. 8.

**2006 - August**

Legend

Class_Name

Water Body

NDVI
<VALUE>
0.9 - 1
0.8 - 0.9
0.7 - 0.8
0.6 - 0.7
0.5 - 0.6
0.4 - 0.5
0.3 - 0.4
0.2 - 0.3
0.1 - 0.2
0 - 0.1
-0 - 0
-0.2 - -0.1
-0.3 - -0.2
-0.4 - -0.3
-0.5 - -0.4
-0.6 - -0.5
-0.7 - -0.6
-0.8 - -0.7
-0.9 - -0.8
-1 - -0.9

(a)

**2006 - August**

Class_Name

Salt Body

Water Body

Soil Salinity Index
<VALUE>
0.95 - 1
0.9 - 0.95
0.85 - 0.9
0.8 - 0.85
0.75 - 0.8
0.7 - 0.75
0.65 - 0.7
0.6 - 0.65
0.55 - 0.6
0.5 - 0.55
0.45 - 0.5
0.4 - 0.45
0.35 - 0.4
0.3 - 0.35
0.25 - 0.3
0.2 - 0.25
0.15 - 0.2
0.1 - 0.15
0.05 - 0.1
0 - 0.05

(b)

**Fig. 9.**

The scatter plot shows NDVI (Pixel value) versus SI (Pixel value) with the fitted curve:

$$y = 3.1209x^2 - 3.38x + 0.9648$$

$$R^2 = 0.8637$$

**Fig. 10.**

[Figure]

**Fig. 11.**

**Fig. 12.**

Figure showing a line plot with "Year" on the x-axis (2003–2013) and "Area of irrigated lands (km²)" on the y-axis (5100–5700).

**Supplement:**

**Analysis of decadal land cover changes and salinization in Urmia Lake Basin using remote sensing techniques**

Yusuf Alizade Govarchin Ghale[1*], Metin Baykara[1], Alper Unal[1]

[1]Climate and Marine Sciences Department, Eurasia Institute of Earth Sciences, Istanbul Technical University, Maslak 34469, Istanbul, Turkey

*Correspondence to: Yusuf Alizade Govarchin Ghale (alizade@itu.edu.tr)

**Abstract.** Urmia Lake, located in the north-west of Iran, is one of the largest hyper-saline lakes in the world. In recent years, most of the Urmia Lake has been rendered to unusable lands. Drought and rapid increase in agricultural activities are the most important reasons behind the shrinkage of the lake. This kind of exploitation with the added salinity from irrigation occurring over time has caused increased soil salinity in the basin leading up to desertification. Soil salinity research are crucial to understand underlying causes and consequences of the drying Urmia Lake. In this study, we use remote sensing technology and image processing techniques to detect spatio-temporal variability of salt body, salt affected lands, and development of irrigated lands to estimate the extend of salinization in terms of spectral response of satellite images for the Urmia Lake Basin from 1975 to 2016. The results of this study indicate that salt and salty soil areas has increased dramatically from 1995 to 2014 and more than 5000 km² of Urmia Lake's water surface area was converted to salt or salty soil bodies during this period. Salinization and desertification progresses are not limited to just dried bottom of the Urmia Lake. Although the area of irrigated lands has increased more than two times during the studied period, soil salinity has increased in regions close to Urmia Lake. This desertification in the basin have potential to be the source of dust storms, which have adverse effects on people's life and climate as well.

**Keywords:** Urmia Lake, Soil Salinity, NDVI, Desertification, Irrigated lands

**1 Introduction**

Soil salinity is an environmental hazard that leads to soil degradation causing agricultural productivity loss, especially in arid and semi-arid regions. In Iran's case, salinity is a major agricultural problem mainly due to the use of low quality saline irrigation water, low rainfall, and high soil surface evaporation. Urmia Lake located in the north-west of Iran, is one of the largest hyper-saline lakes in the world (Hasanzadeh et al., 2012). In recent years, most of the Urmia Lake has been rendered unusable; drought and rapid increase in agricultural activities (i.e., opening up more than 80,000 wells and building more than 50 dams) (Iran Water Resource Management Company, 2016) are the most important reasons behind the shrinkage of Urmia Lake. In a recent study (Fathian et al., 2015), impact of increase in temperature and overuse of water resources on the drying Urmia Lake was investigated and results showed that exploitation of the lake area with the added salinity from irrigation occurring over time has caused increased soil salinity in the basin leading up to desertification. As a results of these environmental changes caused by drying Urmia Lake may lead to spread of diseases, destruction of agricultural lands, and massive damage on economy resulting in mass migration of local people, similar to what has happened in Aral Sea over the past decades (Zarghami, 2011; UNEP, 2012; AghaKouchak et al., 2015).

In a study by Alesheikh et al., (2007), Landsat data and a hybrid technique including band ration and histogram thresholding were used to detect coastline changes in Urmia Lake. In another study, seasonal and annual variations of Urmia Lake region

between 2000 and 2011 were investigated (Sima et al., 2012) by using remotely sensed data. Results of this study showed a decrease of more than 1500 km$^2$ of water surface area of during the 11-year period. In a recent study, Aghakouchak et al., (2015) used Landsat data to determine coastline changes of Urmia Lake between 1972 and 2014. According to the results, the area of Urmia Lake has decreased about 90% starting from 1972 to 2014, which is in agreement with previous research.

5   Kabiri et al., (2012) used Landsat images from 1995 to 2011 to calculate the water surface area of Urmia Lake. Rokni et al., (2014) analyzed coastline changes of Urmia Lake using Landsat data between 2000 and 2013. Hamzehpour et al., (2014) analyzed spatial variation of top soil salinity using groundwater SAR and sampling data on a grid of 500 m in an area of 5000 ha close to Urmia Lake during autumn of 2009 and spring of 2010. Their results indicated inverse correlation between top soil salinity and distance from the lake. Sima and Tajrishy, (2014) used spatial interpolation methods to analyze the

10  spatial heterogeneity and temporal changes of the physiochemical parameters of Urmia Lake between October of 2009 and July of 2010, and found seasonal changes of water quality. Fathian et al., (2015) used three Landsat images to analyze Land use/Land cover (LULC) changes and their impacts on Land Surface Temperature (LST) in the east part of Urmia Lake Basin (ULB) in 1989, 2002, and 2011. Results indicate that human activities and urbanization had modified the vegetation types and LST from the high temperature-sparse vegetation to the low temperature-dense vegetation during the period between

15  1989-2002 and from low temperature-dense vegetation to the high temperature-dense vegetation in period between 2002-2011.

Alizade Govarchin Ghale et al., (2018) have done a detailed study on climatic conditions and main factors behind the drying up of Urmia Lake. They have concluded that anthropogenic factors play a crucial role in the water level fluctuations of Urmia Lake and their impact on the lake is more significant than climatic factors. According to their study results, impact of

20  precipitation and evaporation factors on drying up of Urmia Lake in recent years, 17% and 3%, respectively, and anthropogenic factors account for about 80% of the shrinking Urmia Lake. In previous studies conducted on this basin, there are no detailed studies on long term salinization progress and development of irrigated lands covering all ULB.

Soil salinity research is important to understand the underlying causes and consequences of the drying Urmia Lake. Considering the lack of sampling data and limitations of field survey studies in Urmia Lake Basin, our aim is to use remote

25  sensing technology and image processing techniques to detect temporal and spatial variability of salt body, salt affected lands, and development of irrigated lands to estimate the extend of salinization in terms of spectral response of satellite images for the ULB. Temporal changes of different land cover types including salt, salty soil, and water bodies were determined from 1975 to 2016. Normalized Difference Vegetation Index (NDVI), soil Salinity Index (SI), and Maximum likelihood classification methods were used to quantify the acceleration of salinization in the Urmia Lake Basin for the study

30  period.

**2 Study area**

Urmia Lake has a surface area ranging between 5000 and 6000 km$^2$ and is located in the northwest of Iran (N 37.5°, E 45.5°) between West Azerbaijan and East Azerbaijan provinces with a catchment area of 51876 km². The size of this catchment area is approximately 3% of entire area of Iran. The catchment contains 21 permanent and 39 episodic rivers (Ghaheri and

35  Baghal-Vayjooee, 1999). It is the largest inland lake of Iran and the second largest hypersaline lake in the world (Eimanifar and Mohebbi, 2007; Zarghami, 2011; Hasanzadeh et al., 2012). Saline lakes are susceptible to environmental changes because their properties vary with the changes occur in their hydrologic budgets (Romero and Melack, 1996). The geology of the area ranges from pre-cambrian to Quaternary (Alipour, 2006). By many aspects of chemistry, sediments, and morphology, Urmia Lake is similar to Great Salt Lake in State of Utah in USA. The lake is divided into two parts: northern

and southern parts. These parts are separated by a causeway that has a 1500 m long bridge, which allows little water exchange between two parts (Eimanifar and Mohebbi, 2007; Rezvantalab et al., 2011; Sima and Tajrish, 2013). Its continental climate is affected by the mountains around Urmia Lake and air temperature usually ranges between 0 º C and -20 º C in winter and up to 40 º C in summer. Urmia Lake Basin has an annual average precipitation between 200 mm - 300 mm. (Eimanifar and Mohebbi, 2007). The measured maximum and minimum water surface elevation of Urmia Lake was about 1278.386 m and 1270.168 m in 1995-June and 2014-September, respectively (Iran water resource management company, 2016).

**3 Materials and Methods**

Accumulation of soluble salt in the soil is defined as soil salinity. Remote sensing techniques, either directly or indirectly, are being used to determine soil salinity changes. By using the direct approach, remote sensing techniques applied on terrain surface using crusts or salinity properties of soil. In indirect approach, biophysical characteristics of vegetation types that are affected by salinity are taken into account. Soil salinity can also be identified by electrical conductivity of a solution that is extracted from a water-saturated soil. In agricultural terms, saline soils have an electrical conductivity more than 4 dS (decismens per meter) at $25°C$ (Al-Khaier, 2003). Many salinity indexes are proposed for Landsat images to determine salt-affected soils (Goossens and Alavi Panah, 2001; Al-Khaier, 2003; Abbas and Khan, 2007; Abdul-Qadir and Benni, 2010; Abbas et al., 2013; Allbed and Kumar, 2013; Ahmed and Al-Khafaji, 2014; Arnous and Green, 2015). Among these the two most widely used approaches to monitor soil salinity changes using remote sensing data are Normalized Difference Vegetation Index (NDVI) and Salinity Index (SI).

$$SI = \frac{\rho Green + \rho Red}{2} \tag{1}$$

where, $\rho Green$ and $\rho Red$ are ground surface reflectance values of green and red bands, respectively.

$$NDVI = \frac{\rho NIR - \rho Red}{\rho NIR + \rho Red} \tag{2}$$

where, $\rho NIR$ and $\rho Red$ are ground surface reflectance values of Near Infrared and red bands, respectively.

[revised manuscript text omitted]

has decreased over the years while urban population has increased. The increase in population may have led to the

dramatical increase in agricultural areas.

25

30

**Table 2: Population (million) of both West Azerbaijan and East Azerbaijan**

| Year | Urban Population | Rural Population | Total Population |
|------|------------------|-----------------|------------------|
| 1985 | 2.5 | 2.5 | 5 |
| 1990 | 2.9 | 2.7 | 5.6 |
| 1995 | 3.3 | 2.5 | 5.8 |
| 2005 | 4.1 | 2.4 | 6.5 |
| 2010 | 4.5 | 2.3 | 6.8 |

The northwest part of Iran is one of the agricultural development centers of Iran, and a variety of products, such as apple, wheat, sugar beet, tomato, potato, and grapes are grown in this region. Dams and groundwater resources provide the required water for developing irrigated lands in the basin. There are in total 103 dams in the West Azerbaijan and East Azerbaijan provinces (Iran Water Resource Management Company 2016). Of these dams, 56 are located in Urmia Lake Basin and 42 of them were built between 1990 and 2014. The area of agricultural lands that have been irrigated by dams has been doubled in the last 15 years (Alizade Govarchin Ghale et al., 2018). Ground water resources such as deep wells and semi deep wells are the other sources of irrigation in the study area. The annual discharge water from ground water resources has increased during recent years. The withdrawal water from groundwater resources was about 2156 mcm in 2012 while it was about 702 mcm in 1972. Therefore, dams and wells prevent water from entering the lake and they have a direct effect on the gradual drying of Lake Urmia. As reported in previous studies (Abbas and Khan, 2007, Abdul-Qadir and Benni, 2010; Abbas et al., 2013; Allbed and Kumar, 2013; Arnous and Green, 2015), there exist a strong contrast between normal and salt-affected (salinized) soils in respect to their ground surface conditions. Salt-affected soils have a distinctive feature that make it easier to characterize them; salt efflorescence that had been accumulated over the soil surface by the capillary rise of low quality water. This feature is prominent on top soil and easy to capture in the satellite data. Since salt-affected soils have high spectral reflectance (Metternichet and Zinck, 2003), especially in the blue band of the visible window their spectral response is higher than normal soils. A field study was conducted where over thousands of ground control points were recorded using GPS while photos of these ground control points were analyzed in order to help classify satellite images in this research. Figure 2 shows the salinization and desertification progress as recorded by sample photos over the years. Water body either dried completely (Figure 2 (b)) or withdrawn from coastal parts to center of the lake (Figure 2 (d)). Shriking water body leaves dry, arid, saline soils/ground that increases albedo and affecting the climate of the region. In these pictures, different land cover types such as water, salt, salty soil, and soil bodies are visibly distinguishable.

[Figure]

(a)

(b)

(c)

(d)

[revised manuscript text omitted]

Iran Water Resource Management Company (2016) WRM. http://wrm.ir/. Accessed 15 March 2016

United States Geological Survey (2016) EarthExplorer. https://earthexplorer.usgs.gov/. Accessed 15 August 2016

United States Geological Survey (2016) EarthExplorer. https://glovis.usgs.gov/. Accessed 20 August 2015

35  West Azerbaijan Meteorological Office (2016) http://www.met-ag.ir/. Accessed 10 July 2016

East Azerbaijan Meteorological Office (2016) http://www.eamo.ir/. Accessed 15 July 2016

40